# Clinical Whole-Genome Sequencing Assay for Rapid *Mycobacterium tuberculosis* Complex First-Line Drug Susceptibility Testing and Phylogenetic Relatedness Analysis

**DOI:** 10.3390/microorganisms11102538

**Published:** 2023-10-11

**Authors:** Bennett Shaw, Benjamin von Bredow, Allison Tsan, Omai Garner, Shangxin Yang

**Affiliations:** 1Department of Pathology and Laboratory Medicine, UCLA David Geffen School of Medicine, Los Angeles, CA 90095, USA; bmshaw@mednet.ucla.edu (B.S.); benjamin.vonbredow@corewellhealth.org (B.v.B.); attsan@mednet.ucla.edu (A.T.); ogarner@mednet.ucla.edu (O.G.); 2Department of Pathology, Oakland University William Beaumont School of Medicine, Rochester, MI 48309, USA

**Keywords:** whole-genome sequencing, *Mycobacterium tuberculosis* complex, drug susceptibility testing, phylogenetic relatedness analysis, clinical evaluation

## Abstract

The global rise of drug resistant tuberculosis has highlighted the need for improved diagnostic technologies that provide rapid and reliable drug resistance results. Here, we develop and validate a whole genome sequencing (WGS)-based test for identification of mycobacterium tuberculosis complex (MTB) drug resistance to rifampin, isoniazid, pyrazinamide, ethambutol, and streptomycin. Through comparative analysis of drug resistance results from WGS-based testing and phenotypic drug susceptibility testing (DST) of 38 clinical MTB isolates from patients receiving care in Los Angeles, CA, we found an overall concordance between methods of 97.4% with equivalent performance across culture media. Critically, prospective analysis of 11 isolates showed that WGS-based testing provides results an average of 36 days faster than phenotypic culture-based methods. We showcase the additional benefits of WGS data by investigating a suspected laboratory contamination event and using phylogenetic analysis to search for cryptic local transmission, finding no evidence of community spread amongst our patient population in the past six years. WGS-based testing for MTB drug resistance has the potential to greatly improve diagnosis of drug resistant MTB by accelerating turnaround time while maintaining accuracy and providing additional benefits for infection control, lab safety, and public health applications.

## 1. Introduction

Heralded as the world’s most successful human pathogen, *Mycobacterium tuberculosis complex* (MTB) is one of the world’s leading causes of death. MTB caused an estimated 10.6 million infections and 1.6 million deaths in 2021 alone, and an estimated 1.7 billion people, or roughly 23% of the world’s population is currently infected [1,2]. While effective treatment regimens exist, global implementation and delivery of care is plagued by issues surrounding access, selection of appropriate regimens, and drug supply, all issues that have been exacerbated by the COVID-19 pandemic [3,4]. The resulting frequent incomplete or ineffective treatment has led to a rise in multidrug resistant tuberculosis (MDR-TB) and extensively drug resistant tuberculosis (XDR-TB) [5]. These highly resistant strains account for a significant increase in mortality, especially in hosts with immunosuppression or HIV co-infection [6,7,8]. The rise of drug resistant tuberculosis is an urgent global public health threat, with recent models indicating that the number of people dying from drug resistant TB will nearly double every five years without intervention [9].

Accurate and timely diagnosis of drug resistance remains a significant barrier to effective treatment of tuberculosis. The current methods for identification of drug resistance in MTB rely on culture-based phenotypic drug susceptibility testing (DST). MTB grows very slowly in laboratory conditions, which can delay drug resistance results and therefore proper regimen selection by months following initial detection [10]. While DST is relatively slow, it remains a cost effective and reliable method of clinical DST testing in low-resource settings where MTB is endemic [11]. The introduction of rapid molecular testing, which relies on PCR to provide preliminary antimicrobial resistance (AMR) results for a few drug targets, has already proven that improvements in diagnostic technologies have the potential to change the landscape [12]. Furthermore, several studies have shown that whole-genome sequencing (WGS), which can detect resistance-predicting mutations across the entire genome in under one week, has the potential to improve and accelerate detection of drug resistant TB even further [13,14,15].

The use of WGS for investigation of MTB drug resistance first occurred in September 2006, when an outbreak of XDR-TB in South Africa was identified. Scientists sequenced the genomes of three isolates in one of the first applications of next generation sequencing technology of a microbial pathogen, identifying both known and putative drug resistance mutations [16]. Over the next ten years, the use of WGS in MTB management matured, with Public Health England supporting clinical diagnosis of resistance and outbreak investigations with routine sequencing of MTB as early as 2016 [17]. Since the introduction of standardized, easy-to-use computational analysis tools in 2015 [15], the use of WGS in MTB diagnosis and surveillance has exploded and is now used by public health authorities worldwide. Studies performed in the United States, China, England, Tanzania, France, Indonesia, and Thailand have repeatedly showcased excellent performance and clinical benefits of using WGS in MTB management [13,14,15,18,19,20,21,22]. While accuracy in comparison with phenotypic testing in these studies varies across drugs, the overall concordance is generally very high at >95% [23].

In addition to improvements in diagnostic speed, the data generated by WGS-based testing can be utilized for additional clinical and public health benefit. For example, WGS testing provides an opportunity to conduct molecular surveillance of antibiotic resistance mechanisms. Unlike many other bacteria, where the presence or absence of horizontally transmitted mobile genetic elements encoding antibiotic resistance genes play an important role, drug resistance in MTB can be accurately detected by methods that search for the known single nucleotide polymorphisms (SNPs) present in chromosomal genes like *pncA* and *katG*. Molecular surveillance of antibiotic resistance in MTB therefore primarily revolves around cataloguing SNPs in genes where mutations are known to confer resistance. Beyond surveillance of drug resistance, analysis of genetic relatedness can be used to evaluate suspected transmission events between epidemiologically linked cases, detect laboratory contamination events, and identify mixed infections with high confidence [24,25,26]. Phylogenetic analysis provides the opportunity to monitor the dynamics of local outbreaks in real time by providing insight into circulating lineages, patterns of drug resistance, and geospatial links between isolates [27]. Of particular importance to tuberculosis, surveillance for cryptic transmission between cases with no known epidemiologic links can begin to tease apart the role of active transmission versus reactivation of latent infection as the driving forces of local outbreak dynamics.

To evaluate these benefits, we present the development, validation, and implementation of WGS-based AMR testing of MTB in a clinical laboratory setting in conjunction with both retrospective and prospective evaluation of diagnostic accuracy and clinical utility. Our algorithm leverages TB Profiler, a free and widely available web-based tool [15,28], along with a simple custom quality control algorithm, for rapid and accurate identification of resistance to five commonly used therapeutics (Figure 1). We compare results from WGS-based testing with phenotypic testing, showing high concordance between methods. In the prospective analysis arm of our study, we analyze the improvement in turn-around-time afforded by WGS-based testing. Lastly, we use genomic epidemiology to showcase how genetic data can be used to answer specific clinical questions as well as provide insight into the transmission dynamics of MTB infections in Los Angeles, CA over the past six years.

## 2. Materials and Methods

### 2.1. Ethics

This study was reviewed by the UCLA Human Research Protection Program and received an IRB exemption. All the patients’ information was de-identified.

### 2.2. Clinical Samples

A total of 38 isolates from 38 unique patients receiving care in Los Angeles, California, United States of America were selected for analysis. The isolates selected for analysis are MTB cultures derived from clinical samples that tested positive for MTB in routine clinical testing at UCLA Health Clinical Microbiology Laboratory by Vitek MS MALDI-TOF (Biomerieux, Hazelwood, MO, USA), Gene Xpert PCR (Cepheid, Sunnyvale, CA, USA), or in-house sequence-based tests for pathogen identification [29]. MTB testing in these patients was ordered by clinical treatment teams based upon clinical suspicion for tuberculosis infection. There were no exclusion criteria. Among the 38 patients, 23 were diagnosed with pulmonary tuberculosis, 8 were diagnosed with extrapulmonary tuberculosis without dissemination, and 7 were diagnosed with disseminated, or miliary, tuberculosis. All isolates were collected prior to initiation of treatment with antimicrobial therapy. Additionally, one reference strain, *M. tuberculosis ATCC 21577*, was used for internal procedural validation. An additional 25 biological replicates derived from the 38 clinical isolates were analyzed for repeatability, reproducibility, and culture media cross validation, making up a total of 64 unique sequence data sets included in this study. Of these, 16 samples were isolated from liquid mycobacterial culture medium (MGIT) using the BD BACTEC MGIT Automated Mycobacterial Detection (BD, Franklin Lakes, NJ, USA). The remaining 48 were isolated from solid mycobacterial culture media.

### 2.3. Isolate Preparation, DNA Extraction and WGS

The MTB isolates were heat-inactivated (100 °C for 30 min), and an additional bead-beating step was performed for mechanical disruption of the cell wall. The Qiagen (Valencia, CA, USA) EZ1 Blood and Tissue Kit and the EZ1 Advanced XL instrument were used according to the manufacturer’s instructions to extract genomic DNA from pure microbial isolates. Extracted DNA was quantified with the Qubit 1× double-stranded DNA HS assay using the Qubit 3.0 Fluorometer (Thermo Fisher, Waltham, MA, USA). Acceptable quantities of DNA were ≥0.04 ng/μL. The DNA library was prepared using the Illumina DNA Library Prep kit and WGS was performed on the Illumina MiSeq using the 2 × 250 kit (Illumina, San Diego, CA, USA).

### 2.4. Sequence Data Quality Control Criteria

Using CLC Genomics Workbench 23 (Qiagen, Redwood City, CA, USA), raw sequencing reads were trimmed and mapped to the H37Rv reference genome using a local alignment strategy (NCBI GenBank Assembly GCA_000195955.2). The number of mapped reads for each isolate was recorded. Target region coverage depth was generated by calculating the percentage of nucleotide sites across a custom panel of 15 genes (*ahpC*, *eis*, *embB*, *fabG1*, *gyrA*, *gyrB*, *inhA*, *kasA*, *katG*, *pncA*, *rpoB*, *rpoC*, *rrl*, *rrs*, *Rv0678*) and the neighboring 100 nucleotides in either direction with greater than 15X coverage. This panel includes all described genes associated with resistance to first- and second-line therapeutics and covers 68.2% of all resistance-associated variants present in the TB-Profiler database. To construct this panel, the sequence of all 15 genes along with 100 additional nucleotides in each direction was collated into a target region track for QC analysis in CLC Genomics Workbench. Nucleotide variants were identified using the Basic Variant Detection and InDels and Structural Variants tools. Variants were annotated using Annotate with Overlap Information and Amino Acid Changes tools. Coverage depth and allele frequency were calculated at each variant site.

### 2.5. Mutation Profiling and Resistance Calling Using TB-Profiler

Raw reads in FASTQ format were uploaded to TB Profiler (https://tbdr.lshtm.ac.uk/, accessed on 1 May 2023) for resistance identification. TB Profiler is a web-based application that performs automated genome assembly, variant calling, identification of resistance-associated variants and drug resistance calling [15,28]. Drug resistance results for each sample were recorded along with the reference genome location of all drug resistance supporting variants. Antimicrobial resistance calls from TB Profiler were accepted if the isolate passed quality control and a literature review verified the variant link to antimicrobial resistance. Variants that did not pass literature review were recorded to prevent future false positive results. If TB Profiler was unable to identify any variants associated with antimicrobial resistance and the sample passed quality control, the sample was considered sensitive to all therapeutic agents.

### 2.6. Reproducibility and Repeatability Studies

To assess repeatability, three technical replicates of one randomly selected isolate were prepared and sequenced on the same sequencing run. To assess reproducibility, three technical replicates of three randomly selected unique isolates were prepared and sequenced across three sequencing runs. Computational analysis was then performed to assess concordance between technical replicates for both studies.

### 2.7. Cross-Validation of Culture Media

Three pairs of randomly selected isolates (UCLA-658/UCLA-659, UCLA-737/UCLA-738, UCLA-863/UCLA-864) were cultured in both MGIT and solid agar. Sequencing and analyses were performed as above and concordance between culture media was determined by comparing WGS-based testing results.

### 2.8. Clinical Metadata and Phenotypic DST Results

Clinical metadata and turnaround time data were collected from electronic medical record (EMR) chart review. Date collected, clinical site of isolation, date of positive MTB identification, and date of DST results report were collected for analysis. DST were sent out to commercial reference laboratories and were tested for phenotypic resistance to rifampin (RIF), isoniazid (INH), ethambutol (EMB), streptomycin (STM). DST results were collected for evaluation of concordance with WGS testing. For purposes of comparison, intermediate and resistant results were grouped and treated as resistant.

### 2.9. Prospective Evaluation

After initial evaluation, all isolates identified as MTB in the clinical laboratory over the subsequent six months were collected. Eleven samples (UCLA-1139, UCLA-1140, UCLA-1146, UCLA-1182, UCLA-1142, UCLA-1220, UCLA-1221, UCLA-1222, UCLA-1255, UCLA-1303, UCLA-1304) were evaluated prospectively with both WGS-based testing and phenotypic results. Turnaround time, defined as days from culture positivity to final resistance results, was recorded for both methods.

### 2.10. Genomic Epidemiology

For genomic epidemiology analysis, raw reads were trimmed using Trimmomatic (version 0.36) and mapped using bwa-mem2 (version 2.2.1) [30,31]. Variants were then called using bcftools (version 1.17) [32]. Phylogenetic SNP-distance tree was constructed using augur and visualized with ggtree in R Statistical Software (v4.2.3, R Core Team 2023) [27,33]. For distance tree building, IQTREE with GTR substitution model in augur was used to create a tree structure, then the branch lengths were re-scaled to reflect SNP distance between the reference and each isolate [34]. Lineage data for visualization on the tree was extracted from TB-Profiler results. SNP distances were exported from augur and visually analyzed using MicrobeTrace (Centers for Disease Control and Prevention) [35].

### 2.11. Statistical Analysis

Statistical analysis was performed in R version 4.1.2. Correlation of quality control metrics was performed using binomial logistic regression.

## 3. Results

### 3.1. Development and Testing of Quality Control Criteria

To ensure data quality and consistency of analysis for clinical testing, a quality control algorithm and criteria were established and tested. Across all 64 sequence data sets, the average number of mapped reads was 2,516,285 (235,422–4,826,617) and the average percentage of target regions with coverage greater than 15X was 95.6% (0–100%). We found that samples with greater than 1.5 million mapped reads reliably had coverage depth >15X at greater than 99% of target region sites (Figure 2). Based on this data we determined that samples must meet both greater than 1.5 million mapped reads and greater than 99% of target regions with >15X coverage depth for clinical testing. 57 of 64 sequencing replicates met these quality criteria and were further analyzed using TB-Profiler. To ensure only high-quality variant calls were used to support resistance results, we set quality control criteria for each individual variant identified by TB-Profiler at coverage >15X and allele frequency greater than 80%. Nine unique variants were identified by TB-Profiler, and each met these criteria.

### 3.2. Turnaround Time

During the prospective stage (post-launch validation) of our study, WGS AMR testing was found to decrease turnaround time for all eleven samples evaluated at an average of 36 days faster than phenotypic DST testing. For one sample, WGS-based results were reported 123 days before the DST results were available. The average turnaround time for WGS testing from culture positivity was 23 days while average turnaround time for phenotypic DST testing was 60 days (Figure 3). For two of our samples, WGS testing results were reported before a solid culture isolate was prepared to be sent out to the reference laboratory for phenotypic testing. Notably, WGS testing was not immediately performed when culture turned positive; laboratory confirmation for MTB, communication with the providers and getting permission/approval to perform WGS testing usually took approximately 1 week. Since the WGS test was only performed once weekly, when a sample missed a run, it was postponed to the next week’s run.

### 3.3. Accuracy

WGS-based testing of 38 clinical isolates yielded a total of 13 samples (34.2%) resistant to one or more therapeutics with 8 (21%) resistant to one therapeutic agent. Multidrug resistance was rare, with only five isolates (13.1%) resistant to two or more drugs. Three isolates (7.9%) were found to be resistant to two agents, one isolate (2.6%) was resistant to three agents and 1 (2.6%) resistant to four agents (UCLA-1021), garnering it an MDR-TB classification.

RIF and EMB resistance were identified only in one MDR-TB isolate while INH and PZA resistance were the most common in our population with seven isolates (18.4%) exhibiting resistance to isoniazid and six isolates (15.7%) exhibiting pyrazinamide resistance. Five isolates (13.1%) exhibited resistance to STM. No isolates were found to be resistant to fluoroquinolones by WGS. Resistance results are summarized in Figure 4 and all resistance associated mutations are summarized in Appendix A.

Concordance between resistance results generated using WGS and phenotypic DST, the current gold standard assay, was found to be 97.4%. Results for RIF, PZA, EMB, and STM were 100% concordant. The only discordant isolate, UCLA-797, was found to have false negative INH resistance by WGS. UCLA-658 was identified as resistant to PZA by phenotypic DST but susceptible by WGS with no mutations in the *pnc*A gene. False resistance to PZA by phenotypic methods due to issues with unstable pH during culture is a well-documented issue in the literature [36,37]. In accordance with the WHO technical guide for MTB WGS analysis, we determined that phenotypic resistance to PZA in the absence of mutations should be interpreted as false resistance [38]. UCLA-866 was determined to be resistant to INH by WGS yet susceptible by phenotypic DST. Upon further investigation, literature suggests that mutations in the *ahp*C gene, as identified here, are compensatory for *kat*G loss-of-function mutations and therefore associated with resistance [39]. However, there is no definitive evidence that *ahp*C mutations alone found in isolation are causative of resistance. In the absence of evidence, we disqualified this mutation and corrected the result to susceptible by WGS.

Only one isolate, UCLA-1021, was tested for fluoroquinolone resistance by phenotypic DST and was found to be susceptible in concordance with the WGS result. No other isolates had phenotypic results available for fluoroquinolones and therefore concordance could not be evaluated. The positive predictive value for RIF, INH, PZA, and STM was 100%. EMB positive predictive value could not be evaluated as no samples exhibited resistance. Negative predictive values (NPV) for RIF, PZA, EMB, and STM resistance were 100%. NPV for INH was 96.8% (95% CI: 91–100%).

### 3.4. Precision and MGIT vs. Solid Agar Isolates Cross Validation

WGS-based testing was found to produce 100% (3/3) reproducible and 100% (3/3) repeatable results without variation in results among samples repeated on the same sequencing run or across sequencing runs. WGS-based testing generated 100% (3/3) concordant results from culture in MGIT liquid broth and solid culture media (Appendix A). In addition, one pair of controls consisting of a wild-type strain (UCLA-868) and a PZA resistant strain (UCLA-732) were tested 20 times and generated consistent results.

### 3.5. Genomic Epidemiology

Phylogenetic relatedness analysis revealed that all four global lineages of *Mycobacterium tuberculosis* (lineage 1–4) as well as *Mycobacterium bovis* are represented in our population, with the majority of isolates belonging to lineage 4 (47%) and lineage 2 (26%). A maximum likelihood phylogenetic tree with branch lengths scaled by SNP distance was constructed to determine relatedness (Figure 5). No clear association between resistance status and lineage or clade was noted, yet the lone MDR-TB isolate (UCLA-1021) was identified as belonging to lineage 4.

Pairwise SNP distance analysis revealed that the closest pairwise distance between any two isolates is 96 SNPs, far greater than the 10 SNP threshold that indicates transmission within the past two years [40]. No clear genetic link between either of the two isolates was identified.

### 3.6. Phylogenetic Analysis for Laboratory Contamination Investigation

During the prospective analysis phase of our study, we were able to use genetic data to answer relevant clinical questions posed by the primary treatment team. During the evaluation of the clinical significance of UCLA-1222’s positive culture result, the treatment team inquired if the *M. bovis* isolate could be due to a laboratory contamination event. Although it was felt to be unlikely a contamination event as no other MTB samples were present in the lab at the same time, we were able to use genetic data to confirm that UCLA-1222 was not closely related (minimum pairwise SNP-distance of 412) to any isolate present in the laboratory for the past six years. Thus, we were able to confidently rule out a laboratory contamination event and help confirm the clinical significance of the positive test result.

## 4. Discussion

Our evaluation of the use of WGS in the routine clinical testing for MTB resistance provides convincing evidence that WGS provides a substantial improvement in turnaround time and reliability without sacrificing accuracy. We also show that WGS-based testing is unaffected by switching to liquid culture media, which speeds up the time to positive culture [41]. Taken together, these benefits result in delivery of resistance data to providers an average of greater than one month earlier than phenotypic testing. In practice, patients with drug resistance can be started on the most effective drug combination much earlier, while patients experiencing side effects can be transitioned safely to more tolerable regimens, thereby increasing adherence.

While we found WGS-based testing to be highly accurate and useful in our small study, there remains room for improvement. Our findings highlight a few pertinent issues to address if genetic analysis is to become the diagnostic norm, particularly with PZA and INH resistance results. In the case of PZA, WGS-based testing may indeed be more accurate than phenotypic testing. Further studies are needed to confirm clinical response to PZA in such isolates. Conversely, issues around INH resistance highlight the limitations of the use of an algorithm that only assesses for the presence or absence of mutations in a database. Generating accurate results from this type of model requires meticulous curation of mutations generated from large high-quality datasets. Limiting database size to a small number of high-quality mutations may lead to false negatives while including many lower-quality mutations could lead to false positives. Therefore, database selection should be influenced by resistance burden within the population being tested. For example, a more conservative approach using a small set of high-quality mutations was shown to be highly accurate in a population with only 1.75% incidence of MDR-TB [14]. In populations with higher levels of resistance, a more comprehensive mutation database that includes lower-quality mutations may have better positive predictive value. Further studies are needed to evaluate the interaction between database construction and resistance burden on the accuracy of WGS testing.

Additionally, for complex methods of resistance that are best explained by the interaction between multiple mutations or genes, dichotomous database-dependent algorithms will fall short. Alternative models that can evaluate multiple types of resistance mechanisms, such as those built with machine learning, are promising yet remain in early development [42,43]. However, given the already excellent performance of WGS testing, these models will likely provide marginal gains in accuracy while adding significant computational complexity. Clinical evaluation is needed to determine the benefit of such models in practice.

The other clear benefit to the routine use of WGS is access to rich genetic data that can be used for many applications beyond resistance identification. As we demonstrate in our study, genetic data can be used to investigate epidemiologically linked case clusters or suspected laboratory contamination events. While genetic data is useful in such situations, the most promising benefit of routine use of WGS is the potential for a larger surveillance network. We showcase this ability by using phylogenetic analysis and pairwise SNP distances to conclude that there is no evidence of cryptic local transmission of tuberculosis within patients tested in our healthcare system in Los Angeles, CA in the past six years. While this conclusion should not be generalized to the greater Los Angeles region as we do not capture a representative sample in this study, it is consistent with individual case epidemiologic data. Most patients in our study reported living in a TB endemic country for some amount of time before moving to Los Angeles, leading us to conclude that tuberculosis infections in our patient population are most likely to represent reactivation of latent infection acquired abroad. More sensitive detection of local transmission will require capture of a larger proportion of cases. Therefore, the future utility of larger surveillance networks will likely be facilitated by a combination of an increase in the use of WGS-based testing in clinical laboratories, secure data sharing, and collaboration between clinical laboratories and public health authorities.

Although multidrug resistance rates continue to rise globally, our study confirms that the rate of MDR-TB remains low in Los Angeles, CA. While WGS-based testing is well positioned to replace phenotypic culture-based methods regardless of the patient population, we found that the greatest clinical utility in our population was the strong negative predictive value of WGS-testing coupled with dramatically improved turnaround time. Studies of WGS-based testing in populations enriched with resistant isolates, especially those with high rates of HIV-coinfection, will be critical to evaluating the true clinical impact of WGS for MTB resistance testing. For now, access to WGS technology, especially in low resource settings where MTB is endemic, remains scarce due to the prohibitive cost associated with WGS testing. In this setting, PCR and DST testing continue to be more feasible options [11]. If the roughly equivalent performance seen in this study is replicated in populations with high levels of resistance, the cost effectiveness of widespread adoption of WGS for MTB resistance testing will likely depend upon the clinical impact of faster turnaround. However, in the foreseeable future, with falling costs of sequencing and increasingly easy access to computation analysis tools, WGS is poised to make significant gains in the diagnosis and surveillance of TB.

## Figures and Tables

**Figure 1 microorganisms-11-02538-f001:**
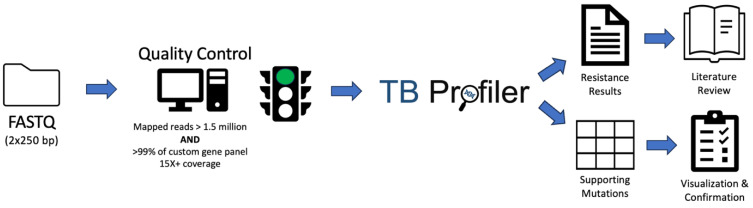
Schematic diagram of computational workflow. Illustrated are the quality control and variant detection steps that lead to resistance results.

**Figure 2 microorganisms-11-02538-f002:**
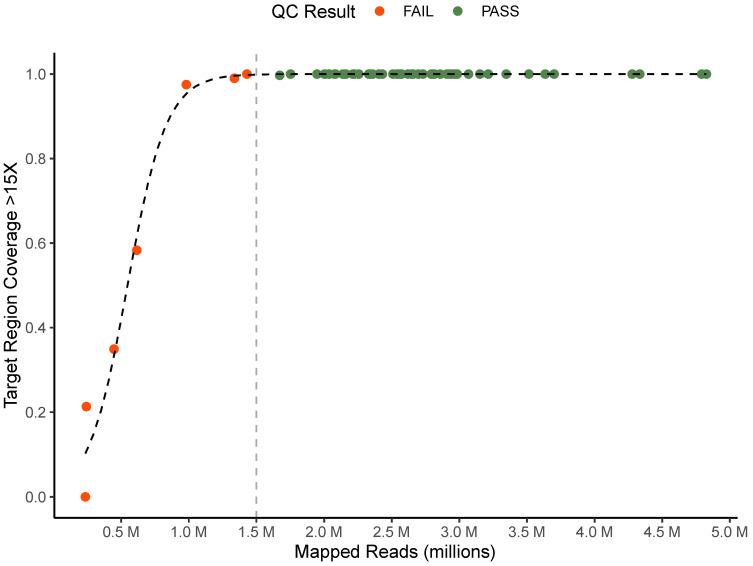
Quality control criteria. Comparison of mapped reads and proportion of target region with greater than 15X coverage for all technical replicates colored by QC status. Vertical line represents mapped read QC cutoff of 1.5 million reads.

**Figure 3 microorganisms-11-02538-f003:**
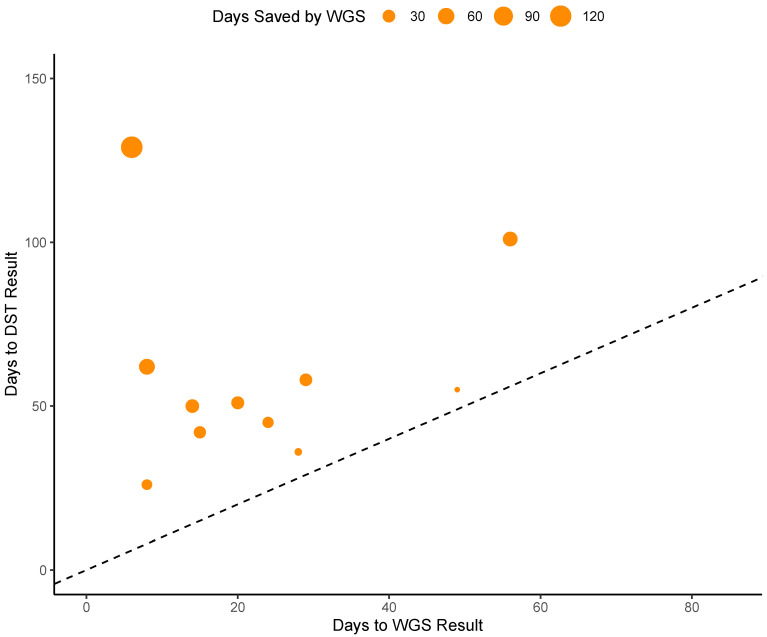
Analysis of times saved by WGS testing. Points above the dashed line represent instances in which WGS testing provided results faster than DST testing. Dot size represents the difference in number of days between WGS and DST turnaround time.

**Figure 4 microorganisms-11-02538-f004:**
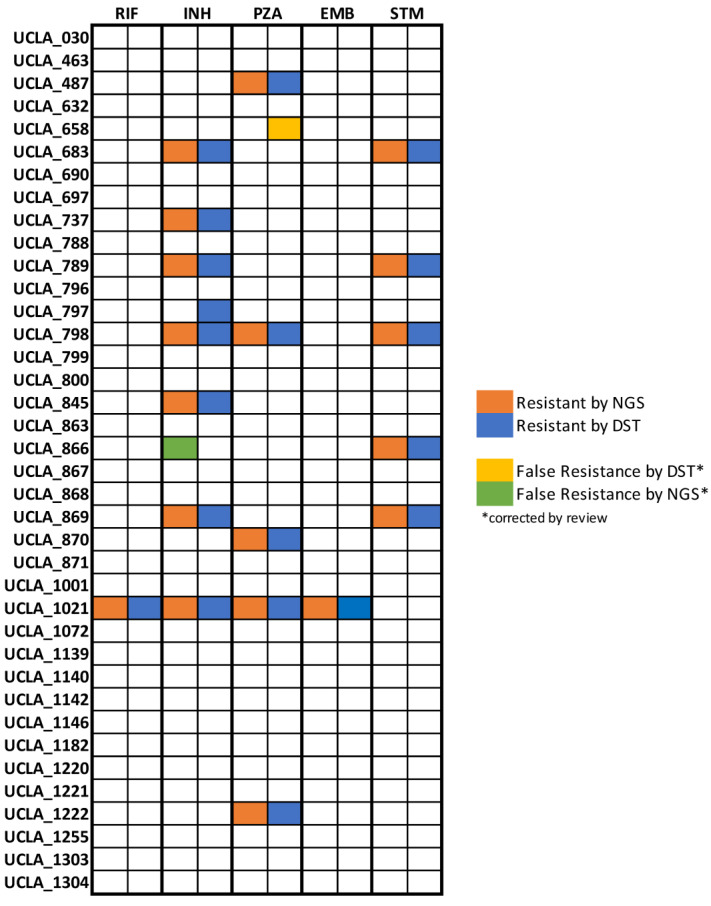
Analysis for concordance between testing methods. Resistance results by drug from whole genome sequencing and phenotypic antibiotic susceptibility testing are displayed for 40 unique clinical isolates. Preliminary results subsequently corrected by literature review quality control are shown for UCLA-658 and UCLA-866.

**Figure 5 microorganisms-11-02538-f005:**
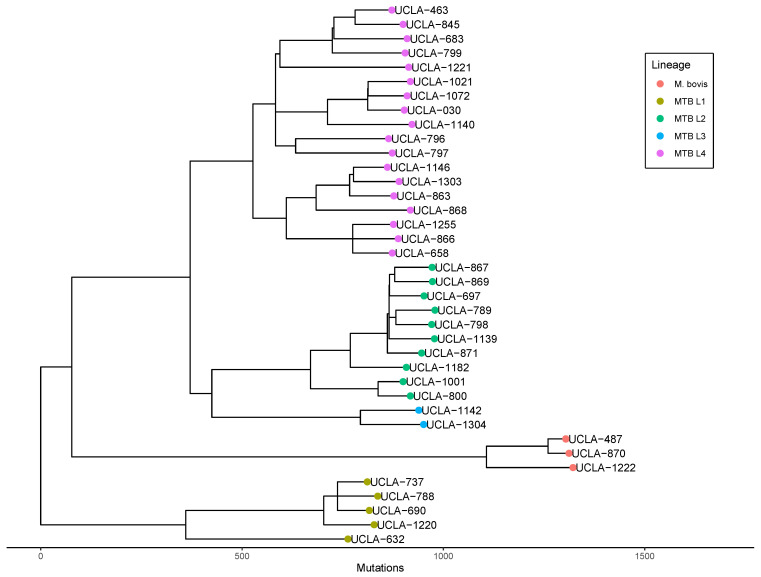
Phylogenetic analysis of MTB in Los Angeles. Maximum likelihood tree with branch lengths scaled by SNP-distance derived from whole genome assemblies of MTB isolates with global lineages displayed by color.

## Data Availability

The data will be available upon request.

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
