# Peer review of "Clinical Whole-Genome Sequencing Assay for Rapid Mycobacterium tuberculosis Complex First-Line Drug Susceptibility Testing and Phylogenetic Relatedness Analysis"

_microorganisms, 2023, doi:10.3390/microorganisms11102538_

Round 1

Reviewer 1 Report

The paper entitled “Clinical whole-genome sequencing assay for rapid Mycobacterium tuberculosis complex first-line drug susceptibility testing and phylogenetic relatedness análisis” is a genomic characterization of antibiotic resistant strains of Mycobacterium tuberculosis over time. However, key methodological aspects are poorly described and leave many doubts about the validity of the analyses presented.

Findings

1.     Abstract section

a.    The origin of the samples must be mentioned. Also, in the abstract indicates that 40 clinical MTB isolates were sampled, but the materials and methods show that 64 samples were analyzed. What is the basis for this difference? these values should be clarified.

b.    The conclusion of this work should be based on the evolution of resistance genes over time.

2.    Introduccion section

a.    a section on the description of resistance genes in MB and their impact on public health should be included.

3.    Materials and Methods section

a.    In the Clinical Samples section, this section should be substantially completed with key points. important features of the sampling process should be included, e.g., criteria for inclusion and/or exclusion of samples and patients or types of infection reported. it is not clear how the retrospective sampling was performed (number and time of samples taken for each patient, e.g. every 3 months). Was the infection evolution   evaluated in each individual? What was the medical treatment during this process? What was the origin of the patients? all these data are of great importance when interpreting and validating the results.

b.    remove underlining in some words

c.     in the Sequence Data Quality Control Criteria section, bioinformatics programs and their respective references should be included.

d.    In the Mutation Profiling and Resistance Calling using TB-Profiler section, cut-off parameters must be included.

e.    In the Genomic Epidemiology, it is not clear how the phylogenetic tree was constructed. key aspects are unknown. what genes were included for the analysis? were they concatenated? what substitution model was used? boostrap?

4.    Results section

a.    the results are presented as a description of the presence of resistance genes vs. their phenotype; but there is no correlation analysis between nucleotide variability over time vs. their variation in antibiotic resistance.

b.    existing mutations in these genes are related to the decrease or increase of resistance? any examples?

c.     were SNPs identified that increase or decrease bacterial resistance?

d.    a genotype vs. phenotype correlation analysis Gene by gene, for example was performed?

e.    How do you interpret the evolution of antibiotic resistance over time? What is the molecular evidence in this regard?

without a deeper analysis in this sense, the point of sequencing the whole genome is lost.

Author Response

Reviewer 1

Comments and Suggestions for Authors

The paper entitled “Clinical whole-genome sequencing assay for rapid Mycobacterium tuberculosis complex first-line drug susceptibility testing and phylogenetic relatedness análisis” is a genomic characterization of antibiotic resistant strains of Mycobacterium tuberculosis over time. However, key methodological aspects are poorly described and leave many doubts about the validity of the analyses presented.

Response: Thank you for your comment. However, the main goal of this study was to evaluate the performance characteristics of using WGS for TB drug susceptibility in the clinical setting. Therefore, we did not present any analysis of evolution of MTB resistance over time as we do not feel that our sampling strategy or sample size is adequate for such an analysis. Please see further explanation below. We instead analyzed the accuracy and speed of WGS-based testing for antimicrobial resistance testing in comparison with current testing modalities and use WGS data to infer relatedness between isolates.

  1. Abstract section
  2. The origin of the samples must be mentioned. Also, in the abstract indicates that 40 clinical MTB isolates were sampled, but the materials and methods show that 64 samples were analyzed. What is the basis for this difference? these values should be clarified.

Response: The abstract text has been updated to reflect the geographic origin of the samples (Los Angeles, CA). Sample counts have been clarified as follows:

There are 38 isolates in this study from unique patients with both WGS-based resistance and phenotypic results. 2 isolates had previously been counted in this total which have now been removed from the analysis. One was reference material from ATCC and the other was an isolate from one of the same patients from a different anatomical site. All concordance values and Figure 4 have been updated to reflect this change. The remaining 24 isolates analyzed are better characterized as biological replicates created for quality control, repeatability, reproducibility, and culture-media cross validation studies. This clarification is now reflected in the material and methods section.

  1. The conclusion of this work should be based on the evolution of resistance genes over time.

Response: Thank you for your comment. However, we have to respectfully disagree with the notion that the conclusion of our work should be based on the evolution of resistance genes over time. Our study is intended to analyze the use of WGS in the clinical microbiology laboratory setting to create actionable data of use to clinical treatment teams and public health authorities. To that end, we analyze WGS as both a diagnostic tool and as a tool for population-level surveillance of transmission dynamics in MTB.

While an analysis of the evolution of resistance genes in such isolates after initiation of antimicrobial therapy would be interesting, our samples represent one time point prior to treatment from 38 unique patients. We do not analyze multiple samples from the same patient, or any samples collected after the initiation of antibiotic therapy. An analysis of the evolution of antibiotic resistance genes over time using the sample collections and study design employed in this study would be inappropriate. Furthermore, in our opinion, analysis of the evolution of antibiotic resistance over time is not often of use to clinical treatment teams or public health authorities and therefore does not fit within the scope of our study.

Additionally, while we could formally analyze resistance patterns over time as they relate to lineage or phylogenetic clade, our small sample size and lack of geographic diversity would limit our conclusions significantly. This type of study would not be feasible with our sample cohort. We have commented on this in the discussion section.

  1. Introduction section
  2. a section on the description of resistance genes in MB and their impact on public health should be included.

Response: Resistance genes, defined as mobile genetic elements that encode antibiotic resistance, do not play a large role in determining resistance in MTB. Therefore, monitoring for the presence/absence of such genes does not play a role in the public health surveillance of drug resistance in the same way that the presence of mecA gene in Staph aureus or NDM-1 in gram negative rods does. A few sentences describing this have been added to the introduction section.

  1. Materials and Methods section
  2. In the Clinical Samples section, this section should be substantially completed with key points. important features of the sampling process should be included, e.g., criteria for inclusion and/or exclusion of samples and patients or types of infection reported. it is not clear how the retrospective sampling was performed (number and time of samples taken for each patient, e.g. every 3 months). Was the infection evolution   evaluated in each individual? What was the medical treatment during this process? What was the origin of the patients? all these data are of great importance when interpreting and validating the results.

Response: More information about clinical samples was added including time range, geographic location, type of infection, inclusion and exclusion criteria, and clarifications as described above. Retrospective sampling was meant to refer to the fact that sequencing of the isolates occurred after initial diagnosis and phenotypic testing was performed. However, this is not the case for all of our samples and therefore the term ‘retrospective’ is inaccurate and has been removed from this section for clarity. Please see response to abstract section comment b for further clarification around the design of this study and the intended analysis.

  1. remove underlining in some words

Response: Underlines were removed.

  1. in the Sequence Data Quality Control Criteria section, bioinformatics programs and their respective references should be included.

Response: The only software used for this analysis was CLC Genomics Workbench, which is already listed and referenced.

  1. In the Mutation Profiling and Resistance Calling using TB-Profiler section, cut-off parameters must be included.

Response: This section was updated to reflect that TB-Profiler was run with default parameters. As TB-Profiler does not report quality control metrics, such as allele frequency or coverage, we created and analyzed our own quality control cutoffs as described in the Results section and summarized in Figure 2.

  1. In the Genomic Epidemiology, it is not clear how the phylogenetic tree was constructed. key aspects are unknown. what genes were included for the analysis? were they concatenated? what substitution model was used? boostrap?

Response: Tree construction using augur was further described. To construct a SNP-distance tree, augur first uses IQ-TREE to build a phylogenetic tree structure using a whole-genome alignment as input, then re-scales branch length using pairwise SNP-distances between the root and each node. All nucleotide sites in the entire reference genome are used to create the tree. As this is a tree based upon SNP distance, no bootstrap values are calculated or displayed. This has been further clarified in the Genomic Epidemiology section of the methods.

  1. Results section
  2. the results are presented as a description of the presence of resistance genes vs. their phenotype; but there is no correlation analysis between nucleotide variability over time vs. their variation in antibiotic resistance.

Response: As described above in response to the comment about the conclusions of our work, such analysis is outside of the scope of this study. This study aims to analyze the accuracy of WGS as a diagnostic tool, in other words as a potential replacement for phenotypic DST testing. We do not attempt to analyze patterns of antimicrobial resistance over time, however we do comment that “No clear association between resistance status and lineage or clade was noted…”

  1. existing mutations in these genes are related to the decrease or increase of resistance? any examples?

Response: Unclear which genes this is in reference to, but mutations in all the genes listed in this study are known to be associated with changes in resistance to antimicrobial therapies. An exhaustive list of these mutations is catalogued in the TB-Profiler database, which we use to determine resistance in this study. Known resistance-conferring SNPs present in the TB-Profiler database that are identified in isolates analyzed in this study are summarized in Supplementary Table 1.   

  1. were SNPs identified that increase or decrease bacterial resistance?

Yes, we identify SNPs that are previously known to increase resistance to the five antimicrobial agents listed in the methods section. All such identified SNPs are summarized in Supplementary Table 1. We do not perform a GWAS analysis that could potentially identify novel SNPs that increase or decrease resistance as such analysis would be extremely underpowered with only 38 isolates, many of which are pan-sensitive.  

  1. a genotype vs. phenotype correlation analysis Gene by gene, for example was performed?

Yes, the central analysis performed in this study is analysis of correlation of the genotype, as defined by the presence/absence of SNPs that confer resistance as identified by TB-Profiler, vs phenotype, as defined by DST testing. The methodology for this analysis is clearly laid out in the methods section and the results are summarized in Figure 4.

  1. How do you interpret the evolution of antibiotic resistance over time? What is the molecular evidence in this regard?

without a deeper analysis in this sense, the point of sequencing the whole genome is lost.

Response: We do not analyze the evolution of antibiotic resistance over time. While we agree that such analysis is important and informative, it is much better achieved with a larger study of isolates with greater diversity across MTB lineage, geography, resistance status and time.

Reviewer 2 Report

Minor comments.

Lines 51-53: Some discussion about the pro-cons versus molecular and genomic procedures, especially when WGS could not be available in endemic poor countries, would be helpful.

Line 53-55:  If the accuracy and data is consistent between methods, saving time (not in all the cases) is cost-worthy? Please add some comment in the discussion

Line 87: Please indicate which reference strain was used.

Line 109: This panel includes all "described" genes

Line 131-132: Sentence typed twice. Please remove.

Line 136-137: randomly selected?

Line 149-150: randomly selected?

Line 182: Figure 1 is not cited in the text

Figure 3 and 4. Is it AST or DST? If it is AST please describe the acronym in the text.

Author Response

Reviewer 2

Comments and Suggestions for Authors

Minor comments.

Lines 51-53: Some discussion about the pro-cons versus molecular and genomic procedures, especially when WGS could not be available in endemic poor countries, would be helpful.

Response: Discussion of this point has been added to the discussion section.

Line 53-55:  If the accuracy and data is consistent between methods, saving time (not in all the cases) is cost-worthy? Please add some comment in the discussion

Response: Discussion of this point has been added to the discussion section.

Line 87: Please indicate which reference strain was used.

Response: This has been added.

Line 109: This panel includes all "described" genes

Response: Corrected in text.

Line 131-132: Sentence typed twice. Please remove.

Response: This typo has been corrected.

Line 136-137: randomly selected?

Response: Yes. This has been clarified in the text.

Line 149-150: randomly selected?

Response: No. All MTB isolates during this period were collected and analyzed prospectively. This has been clarified in the text.

Line 182: Figure 1 is not cited in the text

Response: Figure 1 is now cited in the introduction section.

Figure 3 and 4. Is it AST or DST? If it is AST please describe the acronym in the text.

Response: Corrected to align with the use of DST.

Reviewer 3 Report

In this manuscript, authors reported a developed whole genome sequencing (WGS)-based method for identifying Mycobacterium tuberculosis complex (MTB) drug resistance to rifampin, isoniazid, pyrazinamide, ethambutol, and streptomycin. The algorithm described in this study showed that a rapid and accurate identification of resistance to five commonly used therapeutics was achieved. The plotted data supported the conclusions. Some minor errors can be found, for example, 

1. From Ln 131 “To assess reproducibility” should be revised to “To assess repeatability”?

2. From Figure 2, the “M” following the numbers under x axis could be deleted, as the texts for x axis already shows the “millions”.

3. In Figure 4, the “PZN” on the top of the table should be “PZA”?

Author Response

Reviewer 3

Comments and Suggestions for Authors

In this manuscript, authors reported a developed whole genome sequencing (WGS)-based method for identifying Mycobacterium tuberculosis complex (MTB) drug resistance to rifampin, isoniazid, pyrazinamide, ethambutol, and streptomycin. The algorithm described in this study showed that a rapid and accurate identification of resistance to five commonly used therapeutics was achieved. The plotted data supported the conclusions. Some minor errors can be found, for example, 

  1. From Ln 131 “To assess reproducibility” should be revised to “To assess repeatability”?

Response: Corrected in the text.

  1. From Figure 2, the “M” following the numbers under x axis could be deleted, as the texts for x axis already shows the “millions”.

Response: This has been removed.

  1. In Figure 4, the “PZN” on the top of the table should be “PZA”?

Response: Corrected in Figure 4.